# MESSI: A Multi-Elevation Semantic Segmentation Image Dataset of an Urban Environment

**Barak Pinkovich**                                                    *barakp@campus.technion.ac.il*
*Department of Computer Science*
*Technion-Israel Institute of Technology*

**Boaz Matalon**                                                        *mboaz@technion.ac.il*
*Department of Computer Science*
*Technion-Israel Institute of Technology*

**Ehud Rivlin**                                                         *ehudr@cs.technion.ac.il*
*Department of Computer Science*
*Technion-Israel Institute of Technology*

**Hector Rotstein**                                                     *hector@technion.ac.il*
*Department of Computer Science*
*Technion-Israel Institute of Technology*

**Reviewed on OpenReview:** *https://openreview.net/forum?id=ayWqZ1wyIv&nesting=2&sort=date-desc*

## Abstract

This paper presents a Multi-Elevation Semantic Segmentation Image (MESSI) [1] dataset comprising 2525 images taken by a drone flying over dense urban environments. MESSI is unique in two main features. First, it contains images from various altitudes, allowing us to investigate the effect of depth on semantic segmentation. Second, it includes images taken from several different urban regions (at different altitudes). This is important since the variety covers the visual richness captured by a drone's 3D flight, performing horizontal and vertical maneuvers. MESSI contains images annotated with location, orientation, and the camera's intrinsic parameters and can be used to train a deep neural network for semantic segmentation or other applications of interest (e.g., localization, navigation, and tracking). This paper describes the dataset and provides annotation details. It also explains how semantic segmentation was performed using several neural network models and shows several relevant statistics. MESSI will be published in the public domain to serve as an evaluation benchmark for semantic segmentation using images captured by a drone or similar vehicle flying over a dense urban environment.

## 1 Introduction

In recent years, the interest in using unmanned aerial vehicles (UAVs) or drones to perform various tasks like package delivery, construction site mapping, traffic analysis, structure integrity survey, police search, aerial taxi services, and change detection analysis, has been gaining momentum and attracted the interest of many researchers Mohsan et al. (2023); Han et al. (2022). The successful and safe operation of these tasks on a completely autonomous UAV requires a robust perception of the environment, particularly when sensing is limited to monocular images from a downward-looking camera. Two different but connected research activities have evolved from the motivation above. First, an increasing number of works have appeared focusing on exploring alternatives for semantic segmentation of aerial images using deep neural networks, given the substantial improvement this method offers compared to traditional ones. Second, datasets of

---

[1]The dataset has been published at `https://isl.cs.technion.ac.il/research/messi-dataset/`

drone images are being created to train and benchmark potential semantic segmentation solutions. For instance, the web page *Papers with Code* [2] contains references to 326 datasets when filtered for Semantic Segmentation. Among them, it is possible to find the now classical KITTI Geiger et al. (2012); Liao et al. (2022) or Cityscapes Cordts et al. (2016), which have been extensively used for different learning tasks, including semantic segmentation. Among them, only a few contain aerial views, whereas some are unsuitable for urban scenes, e.g., Gupta et al. (2019); Van Etten et al. (2021).

In addition, although drones enable flexibility in terms of data collection, this same flexibility may pose new challenges to the semantic segmentation task. For instance, the review Han et al. (2022) enumerates some of the challenges arising when using drone images for perception:

1. Uneven object distribution and size, resulting in acquiring images from different altitudes at a constant focal length.

2. Multiple viewpoints and occlusions due to drone mobility in three dimensions. For instance, segmentation may fail to segment objects from the background when objects are far from the camera.

3. Limited power supply and computation hardware if the segmentation scheme is to be implemented on a relatively small platform.

Although these challenges are quite general, they become especially acute when a drone flies over a dense urban environment and altitude changes are dictated by regulatory constraints or exploited to achieve specific objectives. More specifically, and following Han et al. (2022), this paper proposes the MESSI Dataset for evaluating semantic segmentation algorithms as applied to the emergency landing problem as the background application while addressing the problem at different scales (spatial resolution) and multiple viewpoints when collecting images and performing semantic segmentation while flying at different elevations in an urban area as proposed in Pinkovich et al. (2022).

When using deep neural networks to solve the semantic segmentation problem, a sufficiently rich and labeled dataset is needed for training to achieve satisfactory results in real scenarios. The recent work Nigam et al. (2018) considers some of the properties such a database must have to serve as a benchmark for semantic segmentation of drone images. Since UAVs may change their altitude during flight, objects may appear in various sizes. Additionally, UAVs are generally equipped with a wide-angle lens. Thus, objects can be viewed from various angles. Motivated by these observations, the Multi-Elevation Semantic Segmentation Images (MESSI) dataset was constructed using analyzed and labeled aerial images captured by a high-resolution downward-looking camera on a drone flying over a dense urban environment. This paper aims to present MESSI and show how it can be used to train deep neural nets to perform semantic segmentation. To summarize, MESSI's main contributions are:

1. Provide a dataset of images from an urban region captured by a drone at different altitudes. The drone views every area on multiple scales and in different spatial resolutions. The dataset can be used to train a semantic segmentation model that will capture the richness of a 3D flight and thus improve accuracy at inference.

2. Include location and orientation data per image and the camera's intrinsic parameters to extend MESSI to other research topics such as localization, navigation, and tracking (e.g., Chen et al. (2018b)).

3. Enable researchers to improve their autonomous tasks by modeling phenomena such as the effect of how training a network at single or multiple altitudes improves or degrades model performance (see Fig. 7).

## 2   Related Work

The motivation for building the MESSI dataset was to provide data for training and comparing deep-learning semantic segmentation algorithms using images captured by the downward-looking monocular camera of a

---

[2]https://paperswithcode.com/datasets?task=semantic-segmentation

drone flying over a dense urban environment. The number of existing datasets more or less related to MESSI is quite large. The characteristics of the most relevant ones are summarized in Table 1, to guide the reader before delving into a detailed description. For example, openDD Breuer et al. (2020) presents

Table 1: Main characteristics of existing datasets relevant to MESSI

| Name | Features | Main Application | Source |
|------|----------|------------------|--------|
| openDD | Urban | Traffic | Drone |
| DroneDeploy | Unstructured. No overlap. Single altitude | 6 classes segmentation | Satellite |
| Rural Scape | Rural area. Multiple altitude | Video segmentation | HF video |
| Mid-Air | Unstructured. Low altitude | Autonomous flight | HF simulated |
| UAVid | Urban. Single altitude. $45^o$ | 8 classes segmentation | UAV |
| UDD | University. 2 altitudes | 6 classes segmentation | Drone |
| VDD | Urban. 2 altitudes | 7 classes segmentation | Drone |
| ICG | Urban. Low multiple altitudes | 20 classes auonomous flight | Drone |
| AeroSpaces | Single low altitude. No overlap | 12 classes segmentation | Drone |
| CityScapes | Urban. Stereo video | Ground level understanding | Road vehicle |
| ManipalUAVid | University. Video. Single altitude | 4 classes segmentation | Drone |
| MESSI | Urban. Multiple altitudes & revisiting Includes pose | 16 classes segmentation. Search sensor | Drone |

a large collection of drone-captured images in an urban environment, focusing on traffic, and Blaga & Nedevschi (2020) evaluates three aerial datasets: DroneDeploy DroneDeploy (2019), RuralScape Marcu et al. (2020), and Mid-Air Fonder & Van Droogenbroeck (2019). Notice that these three datasets—DroneDeploy, RuralScape, and Mid-Air—deal with *unstructured* environments, and Mid-Air specifically deals with high-fidelity simulated images.

A more relevant dataset is UAVid Lyu et al. (2020), containing 300 images that are densely labeled with eight classes, collected from different areas in the urban scene, and aimed at developing algorithms for scene understanding. In UAVid, the camera captures each area from a single altitude with a slant angle of 45 degrees. Consequently, it fails to capture the size and occlusion issues from observing the same environment from different altitudes. Therefore, studying the effect of different altitudes on semantic segmentation is of interest since it may shed light on the overall behavior of an algorithm.

Two related datasets are UDD Chen et al. (2018b) and VDD Cai et al. (2023), where VDD is a new dataset extended with UDD and new annotations from UAVid. The UDD dataset contains 141 images labeled with six classes taken from 60 to 100 meters. UDD's goal is to use semantic constraints for Structure from Motion (SFM), where only a sample of the images was used for semantic information. VDD contains 400 images labeled with seven classes taken from 50 to 120 meters. MESSI stands out from the latter with its unique feature of organically capturing different scenes from different altitudes, allowing the research community to study the changes in information as a function of altitude and potentially infer from one altitude to another. In contrast, UDD and VDD may contain some images taken from different altitudes, but not by design, so as to allow a systematic study.

The ICG Drone Dataset Mostegel et al. (2019) contains 400 images labeled with 20 classes for training and 200 private images for testing. All images are taken from a bird's eye view at an altitude of 5 to 30 meters above the ground. This low altitude range does not provide substantial spatial resolution or scale variations.

Another well-known dataset is AeroScapes Nigam et al. (2018), a collection inspired by the impact of Cityscapes Cordts et al. (2016), a widely used large-scale dataset consisting of urban street scenes on which the instances of each class appear in various scales. AeroScapes is an aerial semantic segmentation dataset containing 3269 images with 11 classes acquired from 141 video sequences from a varying altitude of 5 to 50 meters. AeroScapes includes several objects, some relevant to the applications we are interested in, some not (e.g., sky, boat). AeroScapes lacks the revisiting from different altitudes, the location and orientation per image, and the camera's intrinsic parameters, which are probably the main features of MESSI.

An additional dataset, ManipalUAVid Girisha et al. (2019), was constructed to evaluate semantic segmentation algorithms. It comprises 667 images captured in six different locations on a closed university campus. Images were taken from an altitude of approximately 25 meters, and four different classes were annotated. Again, this dataset lacks the richness required for performing semantic segmentation in an urban environment.

Finally, additional datasets like Cordts et al. (2016) or Ballesteros et al. (2022) include different viewpoints but are less relevant since the former consists of street-level images, while the latter also uses a DSM of the environment to enrich the available information.

## 3   The MESSI Dataset

MESSI is unique in several aspects when compared to existing databases. It consists of a collection of images captured by a drone at different altitudes. Regions are revisited in multiple scales and spatial resolutions, capturing the visual richness observed by a drone performing horizontal and vertical maneuvers.

The drone was flown in horizontal trajectories in Agamim and Ir Yamim neighborhoods in Netanya and Ha-Medinah Square in Tel Aviv. The Agamim sequences consist of three horizontal trajectory sections (paths A to C), each at four different altitudes: 30, 50, 70, and 100 meters. The number of images was roughly inversely proportional to the altitude as the size of the footprint increased. There is an approximately 75% overlap between consecutive frames. The Ir Yamim sequence consists of a horizontal full-coverage trajectory at the same four altitudes. Compared to Agamim, Ir Yamim has a different scene composition and class population, but some classes have visual resemblances. The Ha-Medinah Square sequence consists of images taken while flying in a horizontal trajectory at a single altitude of 60 meters. This sequence has a considerable domain shift compared to the previous. Table 2 summarizes the number and main characteristics of the images collected for each of the above-mentioned sequences. Samples of the horizontal trajectories are available in Appendix Sec. A.5 Tables 10, 11 where the same area is revisited from a different altitude.

MESSI also includes fifteen vertical trajectory sequences on selected locations in the Agamim neighborhood, consisting of images taken by the drone when descending from 120 to 10 meters, with decreasing overlap between images. Each sequence contains either 100 or 125 images. These sequences partly overlap with Agamim's horizontal trajectory scenarios. Table. 7 in the Appendix Sec. A.2 summarizes the number and main characteristics collected at each location. This rather unusual collection can be potentially used to investigate the correlation between semantic segmentation results at different altitudes. An example of the vertical trajectories is available in Appendix Sec. A.6, Table 12, where Agamim Descend 100_0041 is sampled roughly every 25 meters.

All images were collected using a remotely controlled DJI MAVIC 2 PRO. Due to regulation limitations, the drone was flown at line-of-sight over streets by ALTA INNOVATION[3], a licensed operator as mandated by law. Images were taken by a stabilized downward-looking RGB camera while an RTK INS/GPS algorithm was used to achieve high navigation accuracy. For every image in the dataset, the drone's degrees of freedom (i.e., the location and orientation of the drone and gimbal) are shared in CSV files.

Table 2: Horizontal trajectory: number of images per altitude and flight length per area

| | Number of Images per Flight Altitude | | | | | | Horizontal Flight Length/Coverage |
| | 30 [m] | 50 [m] | 60 [m] | 70 [m] | 100 [m] | Total | |
|---|---|---|---|---|---|---|---|
| Agamin Path A | 40 | 25 | - | 18 | 13 | 96 | 195 [m] |
| Agamin Path B | 51 | 31 | - | 23 | 16 | 121 | 320 [m] |
| Agamin Path C | 45 | 28 | - | 20 | 15 | 108 | 340 [m] |
| Ir Yamim | 166 | 73 | - | 39 | 20 | 298 | 145 [m]×215 [m] |
| Ha-Medinah Square | - | - | 52 | - | - | 52 | 300 [m] |

---

[3]https://alta.team

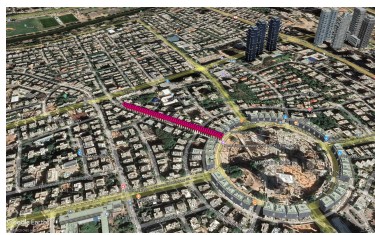 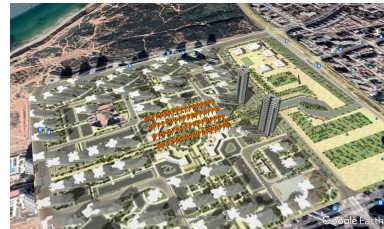 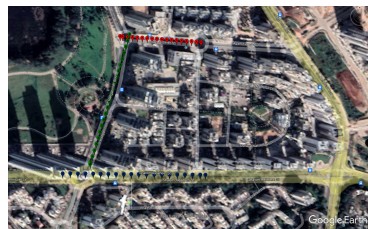

Figure 1: Google Earth views. Left: "Ha-Medinah Square" trajectory (magenta). Middle: "Ir Yamim" coverage trajectory (orange) over the central square. Right: "Agamim" neighborhoods paths A (red), B (green), and C (blue).

The trajectories described above can be combined to train semantic segmentation neural networks in several ways. Fig. 1 shows a Google Earth 3D model and the ground projection of the horizontal trajectories of the areas Ir Yamim and Agamim, in which the images were taken.

### 3.1 Annotations Details

The database comprises 2525 frames with an image resolution of 5472×3684 pixels. Pixel-wise annotations with double-independent checking in our quality control process were done using V7's[4] Darwin automated annotation tool. This tool was chosen for several reasons:

- The tool had a unique feature: polygon tracking. This feature was essential for us because of the horizontal and vertical trajectories. The images in the horizontal trajectories overlap by 50%, so annotations done on the current image can be transferred to the next one while adjusting the mask in the following image with a click of a button. This feature is even more critical in the vertical trajectories, where most objects reappear in the following image but with different scales and orientations. Overall, this feature significantly shortened the annotation process.

- V7 also provided professional subcontractor annotation companies. We defined the object classes, annotation process, and workflow. The workflow and quality control were rigorous. The workflow was defined and monitored online and contained three main phases. The first phase was the annotation. In the second phase, we defined the percentage of mask images that other annotators would check. In the last phase, we defined the percentage of mask images to be checked by us. At each quality control phase, remarks or errors are written to the annotator on the image, and the process goes back to the annotation phase. In fact, we rechecked nearly 100% of the mask images in the third phase. All the quality control was monitored online in the tool.

A class taxonomy table is included in Sec. A.3 of the Appendix. Several relaxation steps were taken without contradicting the overall objective. One such step was annotating buildings as a whole, meaning that any visible objects (e.g., chairs, stairs, objects on the balcony) inside a building were not labeled.

### 3.2 Statistics

A dataset can be partitioned into training, validation, and test sets in many different ways. Since the trajectories of Ir Yamim and Ha-Medinah Square do not overlap with those of Agamim, they were combined to form the test set to evaluate the models' out-of-distribution properties. Test images will be released as an online benchmark with undisclosed ground truth annotations. The *descent* scenarios of Agamim were selected as the training set. In order to limit training time, only one out of five images was taken. Additionally, three *descend* scenarios were excluded from the training set (i.e., 1, 35, and 36) due to their significant similarity with other scenarios.

---

[4]v7labs.com

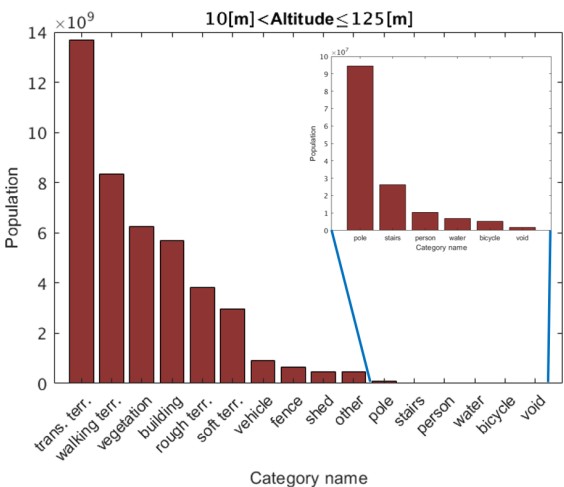

Figure 2: Statistics of the various categories in the combined training and validation sets

Table 3: The selected composition of the training, validation, and test sets

|  | Training set | Validation set | Test set | Images |
|---|---|---|---|---|
| Agamin Descend every fifth image (sets 1, 35, 36 excluded) | + | - | - | 295 |
| Agamin Path A | - | + | - | 96 |
| Agamin Path B | - | + | - | 121 |
| Agamin Path C | - | + | - | 108 |
| Ir Yamim | - | - | + | 298 |
| Ha-Medinah Square | - | - | + | 52 |

Paths A to C were defined as the validation set. Table. 3 summarizes how the dataset was divided. As done in this work, this set may be used for hyperparameter search and can be added later to the training set in order to improve overall results. The class population histogram of the combined training and validation sets is shown in Fig. 2. There is a large imbalance between the highly populated and rare classes (compare *Transportation Terrain* or *Building* with *Person* or *Bicycle*) and can be challenging for a semantic segmentation algorithm, giving poor performance by mostly ignoring some of the rare classes. The next section shows how selecting an appropriate weighting policy can partly overcome this difficulty.

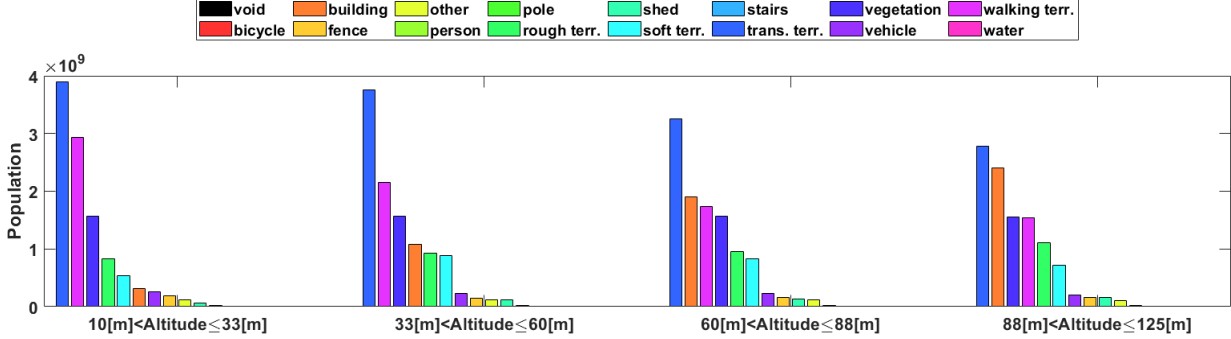

Figure 3: Training and validation set pixel categories statistics per attitude interval

In addition, we investigate the class distribution as a function of altitude by considering four altitude intervals (10, 33], (33, 60], (60, 88], and (88, 125] meters. Fig. 3 shows the four distributions consistent with these

intervals. Notice that the class distribution changes with the altitude intervals, and that three groups of categories change their distribution when the altitude increases. In the first group, the *Transportation terrain* and *Soft terrain* are always the highest and one of the lowest populated classes, respectively, although the distribution within the classes changes as a function of the altitude. In the second group, *Vehicle* and *Pole* are the most and least populated classes, respectively. To further investigate the class population trends, a y-axis log-scaled spaghetti plot is presented in Fig. 4. We can notice, for instance, that the *Building*, *Shed*, *Stairs*, and *Water* populations increase as the altitude interval increases, while conversely, the *Transportation terrain* and *Person* populations decrease.

MESSI enables to study the effect of images taken from different altitudes on semantic segmentation algorithms, a property that is hard to benchmark using existing datasets. MESSI can be used to train the network either on a specific altitude interval or all altitudes, improving the accuracy when testing images with different spatial resolutions taken at different altitudes.

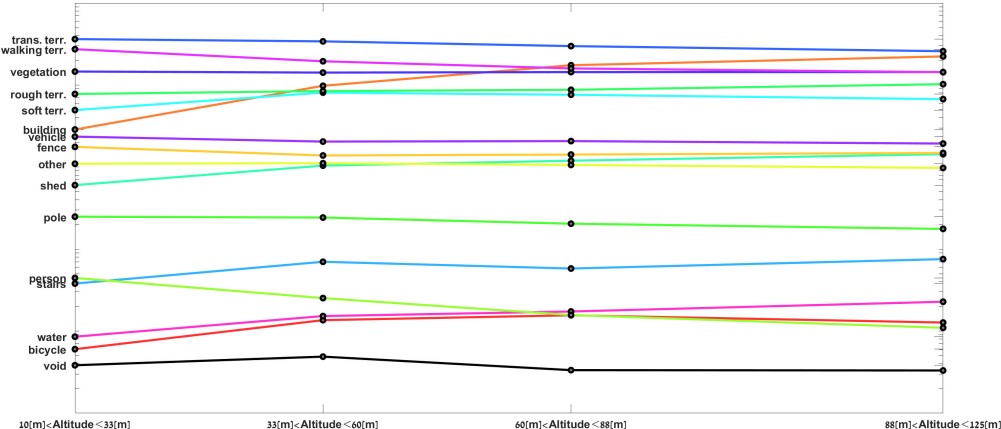

Figure 4: Class over altitude intervals spaghetti plot

# 4 Experiments

This section aims to illustrate how MESSI can be used to train and test a semantic segmentation algorithm.

## 4.1 Implementation Details

The open-source semantic segmentation toolbox MMsegmentation Contributors (2020) was used to train and test various semantic segmentation models. Performance was assessed using the Jaccard index, commonly known as the Intersection-over-Union (IoU) metric, both per category and averaged over all categories (mIoU).

One method of dealing with imbalanced datasets is to increase samples' weight from rare categories. As a result, segmentation accuracy improves in the sense that these categories are not filtered out due to their small representation in the overall set. Three different schemes were tested: 1) uniform weight for all classes (referenced as 'Equal'), 2) a weight inversely proportional to the representation of each class in the training set ('Prop'), and 3) a weight inversely proportional to the square-root of the representation of each class in the training set ('Sqrt'). Note that the class weights are normalized by their total sum to prevent the need to modify some optimization hyper-parameters and the learning rate in particular.

As well-known in the literature, the generalization of a neural net can be enhanced by using a pre-trained model for initializing the optimization process. Since MESSI contains images from multiple heights, the size of an observed object may vary significantly due to its distance from the camera. Hence, an initial model trained on datasets with similar characteristics may prove advantageous. Cityscapes Cordts et al. (2016) is a large and diverse dataset in which instances of each class (e.g., vehicles, poles, buildings) appear in various

scales. For this reason, and similarly to UAVid Lyu et al. (2020), Cityscapes was selected as the dataset for initialization.

Image augmentations during training are another standard way to facilitate generalization. Since a drone can hover at any altitude during the test time, resizing the images may be a good way to improve the segmentation results. Thus, random resizing by up to 15% was employed during training. In addition, a photometric distortion augmentation was performed on the training images to accommodate the model for the slightly changing appearance of surfaces and objects. It includes small random modifications to the images' brightness, saturation, contrast, and hue. Finally, random horizontal flipping was also employed.

The main experiments were performed on an RTX 3090 graphics card with 24 GB memory. Unfortunately, running inference of most medium-sized models (e.g., A.1) on a single 5472×3684 image was found unfeasible, let alone training with such large images. Therefore, and similarly to the approach followed by UAVid Lyu et al. (2020), each image was divided into 3×3 overlapping tiles of 2048 by 1366 pixels, maintaining a ratio of 3:2 as the original image. Two adjacent tiles, either 1712 (horizontally) or 1141 (vertically) pixels apart, should be placed to occupy the entire image. These tiles are fed to the model individually during testing, and the prediction inside overlapped areas is averaged. During training, which requires extra memory due to gradient calculation, each image in the mini-batch (of size two) is a random crop of 1024 by 1024.

Three different main models for semantic segmentation were used on the MESSI dataset. DeepLabV3+ Chen et al. (2018a) is a state-of-the-art convolutional net combining an atrous spatial pyramid that captures multi-scale features with an encoder-decoder structure that gradually refines the boundary between objects. SegFormer Xie et al. (2021) is a relatively efficient Transformer-based segmentation model. Its encoder is a hierarchical Transformer that does not require positional coding, while its lightweight MLP decoder simply aggregates information from layers with different scales. BiSeNetV1 Yu et al. (2018), which was also employed in Pinkovich et al. (2022), fuses a context path with a spatial path. The context allows information from distant pixels to affect a pixel's classification at the cost of reduced spatial resolution. In contrast, the spatial path maintains fine details by limiting the number of down-sampling operations. Two variants from each main model were chosen: the lightest and the heaviest, as long as a batch size of two fits into the GPU memory during training.

Finally, Mask2Former Cheng et al. (2022), a state-of-the-art end-to-end architecture with a key component of masked attention, which extracts localized features by constraining cross-attention within predicted mask regions, was also tested. MMsegmentation does not support the overlapped tiles method in the Mask2Former implementation, so the entire image was needed; therefore, to try to achieve the best results, Mask2Former's largest model whose inference can occupy an A6000 graphics card with 48 GB memory was used. Its training hyper-parameters were set to the same values as all other models here, e.g. each mini-batch (of size two) is a random crop of 1024 by 1024.

The number of training epochs for all models was set at 320, the maximal default value suggested by MMsegmentation. Other than those already mentioned, all hyper-parameters were set to the default values used by MMsegmentation on Cityscapes.

A performance table depicting the running time per crop (and on the entire image for Mask2Former) and memory utilization during inference for all the variants chosen in this paper is presented in the Appendix Sec. A.1, Table 6. As readily observed, BiSeNetV1-18 has the fastest runtime, while SegFormer-B3 and Mask2Former are the slowest.

As mentioned above, MESSI is unique in that it captures areas at different altitudes; thus, another experiment was performed to examine the effect of this feature on performance. Specifically, the SegFormer-B3 model with Sqrt weighting was trained on all "Agamim Path" and "Ir Yamim" scenarios contained within a defined set of altitudes (horizontal trajectories). Then, the model was tested on the entire collection of Descend scenarios (vertical trajectories) without exclusions. All hyper-parameters of the models were chosen as described above, except for the number of epochs that were reduced to 160. The results of these experiments, measured by average accuracy per pixel, are presented in the next section (Fig. 7).

## 4.2 Results

Fig. 5 shows a sample image from Ha-Medinah Square, its ground-truth annotation, and its corresponding prediction using a SegFormer-B3 model. As observed, a large proportion of the image is accurately segmented and classified. The more frequent cases of confusion are between Soft and Rough Terrain, Transportation and Walking Terrain, and Poles and Other Objects.

One of the more noticeable errors appears in the rightmost building, where part of the roof was misclassified as a *Shed*. Although formally an error, it is indeed a shaded area on the roof which, as mentioned in Sec. 3, was filled during the annotation process. Consequently, whenever a building contains objects or surfaces that otherwise should have been annotated, they might be misclassified, as was the case here. This problem can be somewhat relaxed by ignoring the *Building* class both in training and testing and therefore, each experiment was repeated with that category omitted. More sample predictions are available in the Appendix, Sec. A.7, Tables 13, 14.

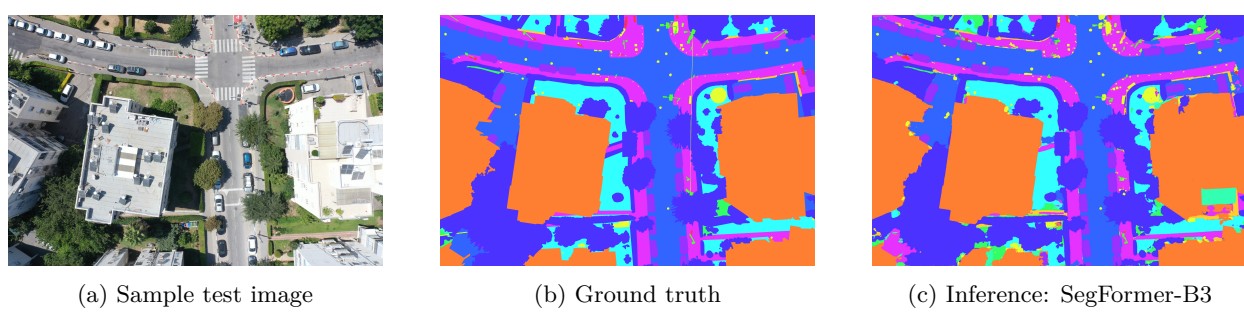

(a) Sample test image      (b) Ground truth      (c) Inference: SegFormer-B3

Figure 5: A visual example of Ha-Medinah Square test set (see table 8 for color legend)

Table 4: Performance on the full test set

| Weighting Method | Model Type | All Categories | | w/o Building | |
|---|---|---|---|---|---|
| | | mAcc | mIoU | mAcc | mIoU |
| Equal | BiSeNetV1-18 | 29.0 | 17.9 | 30.8 | 19.4 |
| | BiSeNetV1-50 | 31.8 | 20.9 | 32.4 | 20.6 |
| | DeepLabV3+18 | 51.9 | 39.8 | 55.6 | 44.4 |
| | DeepLabV3+50 | 57.0 | 45.7 | 59.2 | 47.9 |
| | SegFormer-B0 | 56.8 | 43.7 | 57.6 | 46.7 |
| | SegFormer-B3 | **63.4** | **50.3** | **63.3** | **52.9** |
| | Mask2Former-Swin-B[1] | 57.7 | 45.7 | 57.8 | 38.8 |
| Sqrt | BiSeNetV1-18 | 54.7 | 35.0 | 55.3 | 39.7 |
| | BiSeNetV1-50 | 54.4 | 35.8 | 55.9 | 40.1 |
| | DeepLabV3+18 | 57.5 | 42.7 | 61.3 | 45.7 |
| | DeepLabV3+50 | 59.9 | 44.7 | 62.8 | 48.5 |
| | SegFormer-B0 | 62.5 | 44.1 | 63.8 | 48.8 |
| | SegFormer-B3 | **67.7** | **52.6** | **66.6** | **54.5** |
| | Mask2Former-Swin-B[1] | 57.1 | 44.5 | 57.3 | 40.4 |
| Prop | BiSeNetV1-18 | 27.1 | 6.6 | 30.8 | 7.1 |
| | BiSeNetV1-50 | 20.0 | 3.3 | 21.5 | 4.0 |
| | DeepLabV3+18 | 58.8 | 37.1 | 61.6 | 43.1 |
| | DeepLabV3+50 | 60.9 | 41.4 | 63.2 | 43.8 |
| | SegFormer-B0 | 63.0 | 40.1 | 65.1 | 44.8 |
| | SegFormer-B3 | **64.8** | **47.9** | **67.7** | **51.8** |
| | Mask2Former-Swin-B[1] | 48.0 | 35.8 | 60.9 | 40.7 |

[1] inference on the entire image.

Table 4 shows the performance of the various models on the entire test set. Although the principal measure is mIoU (defined in the previous subsection), mAcc (mean Accuracy) is also presented for completeness. The best result for each of the six selections is boldfaced. Note that weighting methods experiments are not normally conducted when publishing a new dataset (cf. Cordts et al. (2016), Lyu et al. (2020), Nigam et al. (2018) and more). The table shows that: 1) as could be expected, the larger the model, the better it performs; 2) the Sqrt weighting method usually outperforms the others, especially for the least successful models; and 3) almost unanimously, discarding the Building category improves the quantitative results.

Table 5: IoU per category on the full test set when using SegFormer-B3 model

| Weighting Method | Equal | | Sqrt | | Prop | |
|---|---|---|---|---|---|---|
| Inc. Building? | Yes | No | Yes | No | Yes | No |
| building | 56.1 | - | **61.2** | - | 55.7 | - |
| trans. terr. | 74.3 | 74.8 | **76.7** | 74.8 | 71.6 | 71.6 |
| walking terr. | 80.7 | 83.9 | 82.2 | **84.2** | 80.2 | 82.1 |
| stairs | **41.5** | 26.3 | 41.2 | 30.1 | 25.4 | 33.6 |
| soft terr. | 71.0 | 86.8 | 70.9 | **88.1** | 69.2 | 86.0 |
| rough terr. | 44.2 | 39.1 | **44.9** | 43.7 | 42.6 | 43.6 |
| vegetation | 76.9 | 77.3 | 76.9 | **77.5** | 76.9 | 77.4 |
| water | 0.3 | 6.3 | 2.1 | 1.8 | 0.7 | **9.7** |
| pole | 19.7 | 22.0 | 24.7 | **26.8** | 23.5 | 21.7 |
| shed | 51.2 | 64.7 | 63.1 | **71.1** | 52.3 | 67.1 |
| fence | 52.1 | 58.4 | 54.0 | **59.2** | 49.0 | 53.2 |
| vehicle | 90.6 | **91.1** | 89.5 | 90.8 | 86.5 | 87.2 |
| bicycle | 19.2 | 23.5 | 26.5 | **27.5** | 21.7 | 18.5 |
| person | 46.5 | **46.8** | 42.8 | 45.1 | 32.9 | 34.3 |
| other object | 30.3 | 39.8 | 31.7 | **42.5** | 30.3 | 38.5 |
| mIoU with water | 50.3 | 52.9 | 52.6 | **54.5** | 47.9 | 51.8 |
| mIoU without water | 53.9 | 56.5 | 56.2 | **58.6** | 51.3 | 55.0 |

Mask2Former did not perform as anticipated. As mentioned in Sec. 4.1, the training conditions were the same as those for the other models (except for using a GPU with a larger memory size to train the second-largest model). However, due to MMsegmentation's implementation, the inference was performed on the entire image and not on overlapped tiles, which was thought to achieve better results. Mask2Former is region-based, which could explain why it did not perform as well as SegFormer on small objects. Similar results were reported in Wang et al. (2023), where Mask2Former was trained and tested on aerial images. The IoU per category for Mask2Former is presented in the Appendix Sec. A.4, Table 9, where evidently it performed significantly worse on classes such as "bicycle" and "stairs" than SegFormer-B3 (see below).

The IoU per category for SegFormer-B3 (the best-performing model) is presented in Table 5, where the highest score for each category is boldfaced. One can observe that, for the most part, the trends of the overall mIoU are maintained here - using the Sqrt weighting method and discarding the Building category improve performance. Another observation is that the rare categories (e.g., *Water*, *Bicycle*, *Person*) generally get the worst scores. Notable exceptions are *Shed* and *Vehicle*, where the latter actually has the best score among all categories. A possible explanation is that the Vehicle category is quite distinct in appearance.

Another way to interpret the results is by looking at the confusion matrix in Fig. 6. This matrix was calculated for the SegFormer-B3 model (with Sqrt weighting and for all categories). As readily seen, the least populated categories (e.g., *Water*, *Pole*, *Bicycle*, *Stairs*) have significant off-diagonal components. In particular, for the *Water* category, some of its instances are located on buildings' roofs, which explains the significant confusion between them and why it was omitted from Table 5 and from the mIoU calculations. In addition, Pole is often misclassified as Other Objects since low-enough poles were labeled as the latter. Another observation is that the Building category is occasionally confused with Soft Terrain. This stems

from having some unusual buildings with grass-covered rooftops in the Ir Yamim images. Finally, Rough Terrain is frequently misclassified as Vegetation since, looking from above, it may be difficult to distinguish between the two.

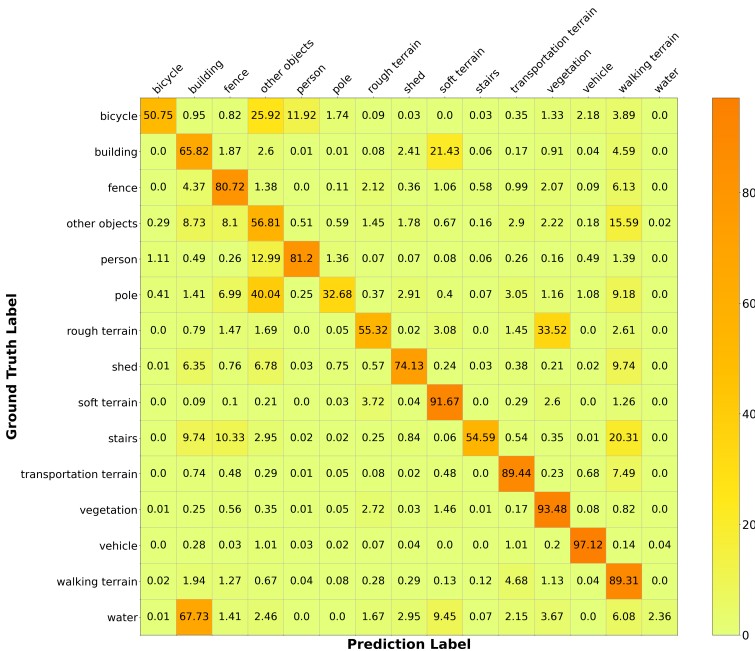

Figure 6: Normalized confusion matrix on the test set for the SegFormer-B3 (All categories, Sqrt weighting)

MESSI was mainly constructed to enable the study of the effect of images taken from different altitudes on semantic segmentation algorithms, a property that is hard to benchmark using previous datasets. Fig. 7 stresses the significant comparative advantage of using MESSI. The figure shows how training the best-performing model SegFormer-B3 with Sqrt weighting scheme on a horizontal trajectory at different altitudes (i.e., "Agamim path" and "Ir Yamim" data) improves or degrades the accuracy when testing on images from the vertical trajectories (i.e., "Agamim descend" data). Fig. 7a shows the accuracy degradation when training at higher altitudes but descending to lower altitudes where no training was performed. For instance, training on images from all "path" altitudes (i.e., 30, 50, 70, 100 meters) has little or no effect on the "descend" test accuracy. However, the average accuracy when training on a limited set of altitudes (like 100 meters in Fig. 7a) results in lower performance as the elevation decreases. Fig. 7b also shows the accuracy degradation, but this time, the training images are taken from lower altitudes, and the drone ascends to higher altitudes where no training was performed. At first glance, training on lower altitudes and extrapolating to higher altitudes seems better. However, this precisely emphasizes the tradeoff:

1. Lower altitude: higher spatial resolution, smaller ground footprint, longer trajectories, more obstacles

2. Higher altitude: lower spatial resolution, bigger ground footprint, shorter trajectories, fewer obstacles

It is worth stressing that accuracy was calculated on the overlapping field of view from all altitudes in each trajectory of the "descend" data area: whereas at the lowest altitude, the accuracy was calculated on the entire image, as ascending, a smaller area of the image participated in the calculation.

## 5   Summary and Conclusions

This paper presents a new dataset called "MESSI," designed to serve as an evaluation benchmark for elevation-based semantic segmentation algorithms. MESSI consists of a rich collection of aerial images at different

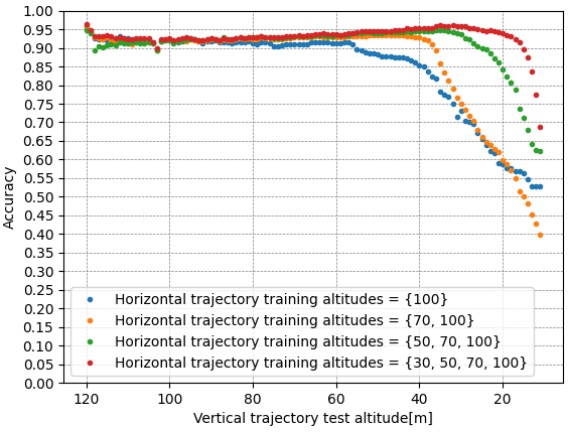

(a) Training with descending horizontal trajectories

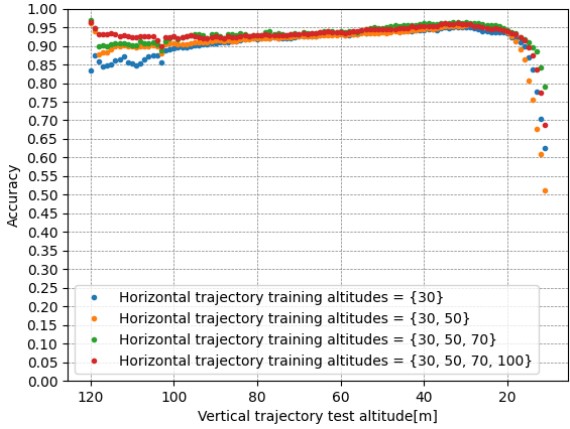

(b) Training with ascending horizontal trajectories

Figure 7: Training altitude impact on semantic segmentation accuracy

altitudes over a dense urban environment and is unique in several aspects. The dataset was annotated using Darwin, the V7 pixel-wise tool, with 15 different classes. Extensive effort was made to make the dataset error-free and consistent. A statistical study of the resulting semantic maps shows that there is a large variability in population between classes, a challenge for current semantic segmentation algorithms.

MMSegmentation Contributors (2020) was used to train and test various semantic segmentation models, and an Intersection-over-Union metric, both per category and averaged, was used to assess performance. Three different weighting policies were used to overcome the population imbalance. Overall, SegFormer MiT-B3 provided the best results over six tested alternatives using three weighting methods.

The results in the paper show that MESSI can indeed be used to train a deep neural network for performing semantic segmentation on aerial images over a dense urban environment. The fact that MESSI contains annotated images from both horizontal and vertical altitudes can be used to study the effect of altitude on semantic segmentation algorithms and properties that are hard to benchmark using existing datasets. Finally, MESSI is published in the public domain to help improve the result presented in this paper, by potentially utilizing other network structures or weighting policies.

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

# A  Appendix

## A.1  Inference Time and Memory Usage

Table 6 presents the running time and memory utilization during inference for all the variants chosen in this paper. The performance for all models is measured per crop, while for Mask2Former on the entire image. As can be seen, Mask2Former is the largest and slowest model.

Table 6: Inference time and memory usage for the different models

| Model Type | Encoder Backbone | Inf. Time (ms./crop) | Inf. Memory (GB) |
|---|---|---|---|
| BiSeNetV1 | ResNet-18 | 8.6 | 6.07 |
| BiSeNetV1 | ResNet-50 | 87.5 | 6.65 |
| DeepLabV3+ | ResNet-18 | 19.8 | 5.31 |
| DeepLabV3+ | ResNet-50 | 160.1 | 7.97 |
| SegFormer | MiT-B0 | 104.9 | 10.5 |
| SegFormer | MiT-B3 | 316.5 | 10.76 |
| Mask2Former[1] | Swin-B | 1550 | 31.317 |

[1] inference on the entire image.

## A.2  Vertical Trajectories Scenarios

MESSI includes 15 vertical trajectory sequences in selected locations in the Agamim neighborhood, consisting of images taken by the drone when descending from 120 to 10 meters, with decreasing overlap between images. Each sequence contains either 100 or 125 images.

Table 7: Vertical trajectory: number of images per location and flight length per area

| Agamim Location | Number of Images per Flight Altitude | Vertical Flight Length |
|---|---|---|
| 100_0001 | 125 | 106 [m] |
| 100_0002 | 125 | 107 [m] |
| 100_0003 | 100 | 97 [m] |
| 100_0004 | 125 | 101 [m] |
| 100_0005 | 125 | 104 [m] |
| 100_0006 | 125 | 107 [m] |
| 100_0031 | 125 | 110 [m] |
| 100_0035 | 125 | 104 [m] |
| 100_0036 | 125 | 108 [m] |
| 100_0037 | 125 | 106 [m] |
| 100_0038 | 125 | 108 [m] |
| 100_0040 | 125 | 103 [m] |
| 100_0041 | 125 | 106 [m] |
| 100_0042 | 125 | 101 [m] |
| 100_0043 | 125 | 101 [m] |

### A.3 Segmentation Taxonomy

The segmentation taxonomy was designed for finding safe, obstacle-free ground areas suitable for landing while considering human, property, and drone safety. Table 8 shows the selected class taxonomy used in MESSI.

Table 8: Classes with brief description

| Class Name | Description | GT Color |
|---|---|---|
| building | without including their ground-floor yards | |
| transportation terrain | road, parking lot, bicycle lanes | |
| walking terrain | sidewalk, walking lanes, playground, basketball court, concrete, deck, porch, etc. | |
| stairs | stairs | |
| soft terrain | grass, soil (with no bushes), gravel, yard, etc. | |
| rough terrain | soil mixed with low bushes, rocks, junk, construction materials, and bleachers | |
| vegetation | bushes, trees | |
| water | any water deposit, including pools | |
| pole | lighting poles (inc. base), traffic signs, etc. | |
| shed | shed, gazebo, sun shades, shaded bus stations | |
| fence | fence, wall | |
| vehicle | any vehicle with four or more wheels | |
| bicycle | bicycles, motorbikes, scooters and other small vehicles | |
| person | either walking or standing (not on or in a vehicle) | |
| other object | including trash bins, benches, phone booths, statues | |
| void | unlabeled pixels (not an actual class) | |

### A.4 IoU Per category on The Full Test Set When Using Mask2Former Model

Table 5 presents the IoU per category on the complete test set when using the SegFormer-B3 model. A similar table containing the IoU per category on the Mask2Former model is presented in Table 9. Please note that Mask2Former performs significantly worse in classes such as "bicycle" and "stairs."

Table 9: IoU per category on the full test set when using Mask2Former model

| Weighting Method | Equal | | Sqrt | | Prop | |
|---|---|---|---|---|---|---|
| Inc. Building? | Yes | No | Yes | No | Yes | No |
| building | 44.45 | - | **49.58** | - | 28.59 | - |
| trans. terr. | **79.81** | 71.35 | 77.15 | 66.72 | 55.06 | 74.69 |
| walking terr. | **84.16** | 82.41 | 79.61 | 76.14 | 70.79 | 79.39 |
| stairs | **5.21** | 1.04 | 0.34 | 0.17 | 2.5 | 2.39 |
| soft terr. | **69.48** | 69.45 | 67.74 | 60.33 | 67.42 | 61.06 |
| rough terr. | **41.33** | 37.97 | 39.82 | 26.1 | 28.84 | 27.64 |
| vegetation | **76.05** | 75.94 | 75.64 | 75.16 | 72.63 | 63.87 |
| water | 0 | **0** | 0 | 0 | 0 | 0 |
| pole | **30.31** | 28.1 | 24.49 | 27.49 | 25.1 | 29.29 |
| shed | 31.38 | 26.18 | 27.86 | **35.32** | 16.19 | 38.18 |
| fence | 48.24 | 46.19 | **50.35** | 45.04 | 47.97 | 35.65 |
| vehicle | **91.26** | 27.54 | 87.83 | 72.96 | 88.03 | 77.96 |
| bicycle | 9.89 | 0.71 | 2.66 | 0 | 0.6 | **12.4** |
| person | 41.39 | 44.75 | **52.71** | 50.73 | 4.84 | 39.56 |
| other object | **32.25** | 31.92 | 31.34 | 29.14 | 28.65 | 27.83 |
| mIoU with water | **45.7** | 38.8 | 44.5 | 40.4 | 35.8 | 40.7 |
| mIoU without water | **48.94** | 41.81 | 47.65 | 43.48 | 38.37 | 43.84 |

### A.5  Sample Images of Horizontal Trajectories

A sample of images of horizontal trajectories is presented in Tables 10, 11. The trajectory is performed at different elevations (30, 50, 70, and 100 meters) in Ir Yamim and Agamim Path A, B, and C. Out of the entire path, a single area has been selected to emphasize the difference in information captured at different altitudes. In contrast, Ha-Medinah Square of the test set is captured only at 60 meters; therefore, various areas of the trajectory are presented.

Table 10: Horizontal trajectories - sampling the same location at all elevations

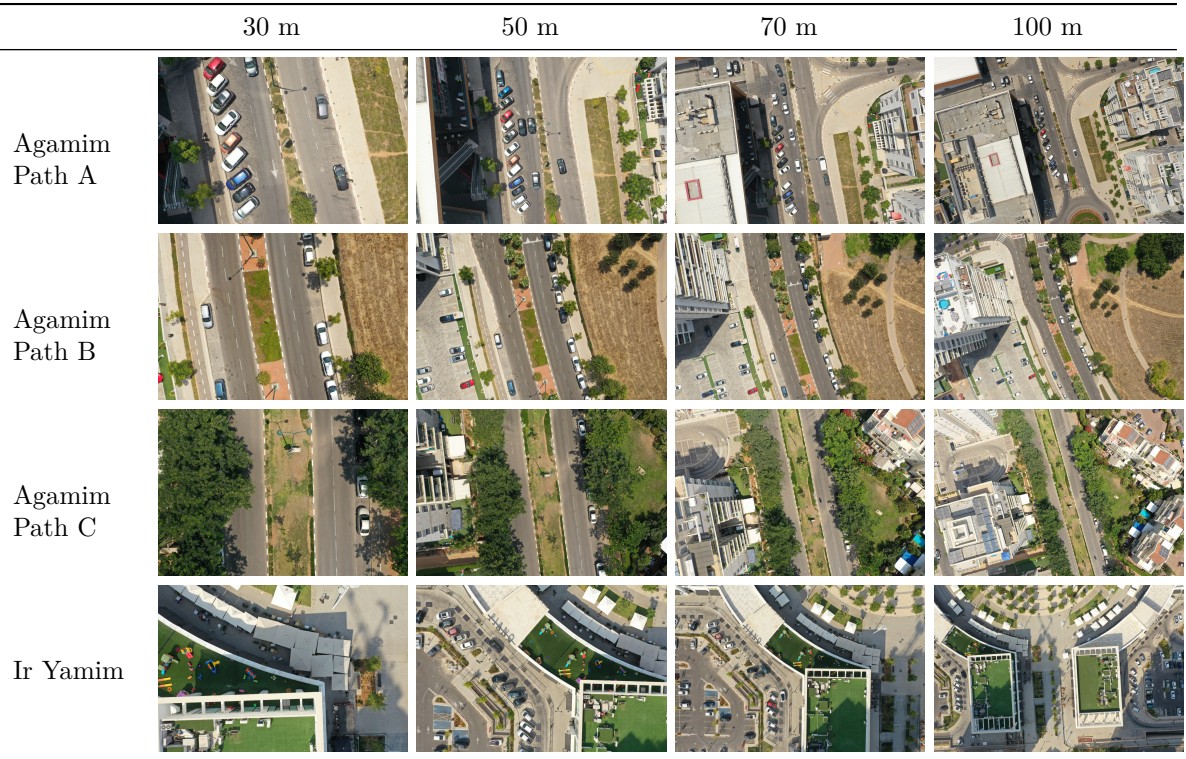

Table 11: Ha-Medinah Square horizontal trajectory - sampling the trajectory at 60 meters

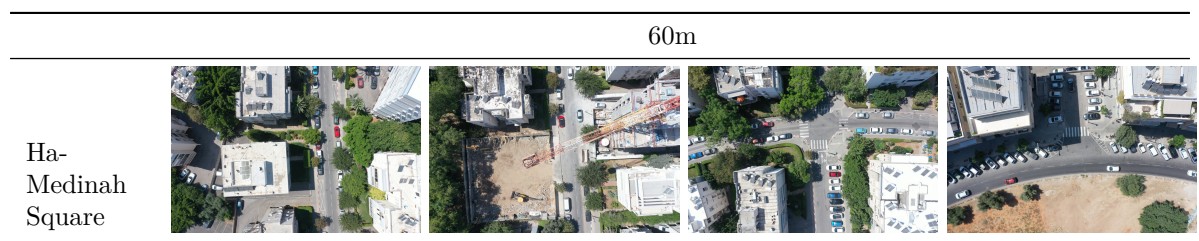

## A.6 Sample Images Of Vertical Trajectories

Agamim Descend 100_0041 is presented in Table 12, in which the drone descends vertically from 125 to 10 meters. The images are sampled roughly every 25 meters.

Table 12: Agamim vertical trajectories - sampling Agamim Descend 100_0041

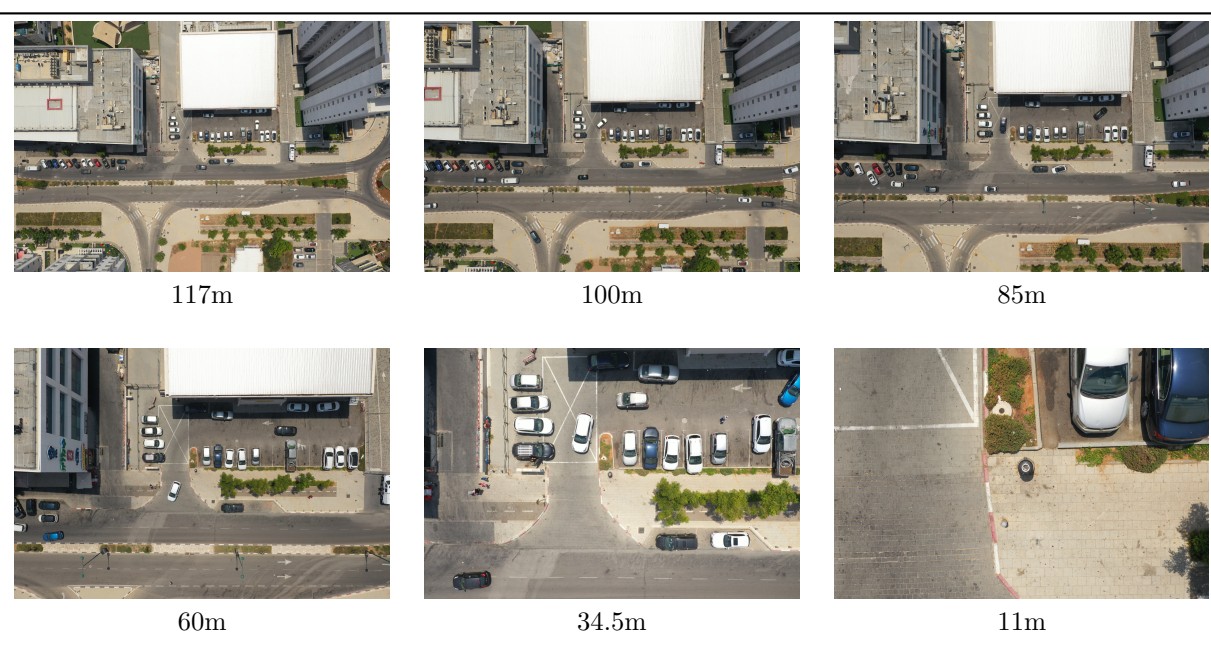

### A.7 Sample prediction images with SegFormer MIT-B3

Some Ir Yamim and Ha-Medinah Square prediction examples using SegFormer MIT-B3 are presented in Tables 13 and 14, respectively. In Ir Yamim, although the "soft terrain" and "other objects" on the roof of the building were not annotated in the ground truth, the model actually predicted them. Moreover, in both Ir yamim and Ha-Medinah Square, the model occasionally confuses between "soft terrain" and "rough terrain".

Table 13: Ir Yamim: Prediction examples with SegFormer MIT-B3

Table 14: Ha-Medinah Square: Prediction examples with SegFormer MIT-B3

