# OpenReview forum: "MESSI: A Multi-Elevation Semantic Segmentation Image Dataset of an Urban Environment"
_TMLR — Accepted by TMLR_

### Review · Reviewer_9e8q · 2024-12-29

**Summary Of Contributions:**

The authors propose MESSI: a dataset composed of drone images of urban environments captured at varying altitudes with semantic segmentation labels. The dataset includes metadata for how and where the data was captured. MESSI captures data from a larger variety of altitudes than previous datasets. The dataset is split into training and validation sets that share a location and a test set composed of two different locations to out-of-distribution generalization. Experiments utilizing MMsegmentation evaluate 7 model architectures across 3 class weighting schemes. Performance across classes is investigated for the best performing model. The effect of altitude variety on model performance is also examined.

**Audience:**

Yes

**Broader Impact Concerns:**

This dataset is clearly relevant to military interests, but I do not have any immediate ethical concerns with this work.

**Claims And Evidence:**

Yes

**Requested Changes:**

R1: Remove the new lines in the OpenReview abstract. (strengthen the work)
R2: The text above Table 3 is missing a space before “and 3)”. (strengthen the work)
R3: Highlight Figure 7 in your third main contribution in the Introduction. (strengthen the work)
R4: Code for loading the dataset and executing experiments should be shared. (critical for recommendation)
R5: The model and class weighting scheme used for Figure 7 should be stated. (critical for recommendation)

**Strengths And Weaknesses:**

**Strengths**

S1: MESSI provides images from a drone at a larger variety of altitudes than previous datasets and includes rich metadata about how the data was collected, such as camera intrinsics.

S2: The discussion of related works is extensive and detailed. As someone who is not familiar with these datasets, it gave me insight into exactly how MESSI is novel compared to previous work.

S3: There are extensive experiments utilizing an existing semantic segmentation benchmarking tool. Multiple metrics and results across class weighting schemes are reported along with a detailed investigation of the behavior of the best performing model.

S4: Figure 7 provides a unique insight into the effects of altitude on model performance.

**Weaknesses**

W1: The description of the annotation process lacks detail. Were annotations fully generated by a deep learning model then checked by human annotators? Do the authors have any relationship to V7 that should be disclosed?

W2: Section 3 focuses on horizontal trajectories, lacking a table like Table 1 for the vertical trajectories. Despite this, the vertical trajectories are used for the training set. How many images are in the training set and is that currently listed in the paper?

W3: Is there any concern about data distribution shift due to using vertical trajectories for train and horizontal trajectories for validation and test that should be discussed? Was there a motivation for this shift?

W4: I interpret the last paragraph of Section 4.1 as saying that the Segformer-B3 model used a different training and test set (what happened to validation?) than the other models. What motivated comparing across splits with different model architectures rather than a separate ablation where the same architecture is trained with different splits?

W5: I interpret the second paragraph of Section 4.2 as saying that it is fine for the model to label the shaded part of the roof as a shed, but do not understand why. On a related note, does the dataset contain inaccurate labels? If so, this should be included in Section 3.

---

> ### Author Response · Authors · 2024-12-31
> **Answer to Reviewer 9e8q**
>
> Dear Reviewer 9e8q29:
> Thank you for your comprehensive and in-depth review. In particular, thank you for appreciating the importance and unique characteristics of MESSI. We have uploaded a revised version of the paper according to your remarks, and we believe that our next answers will help clarify your concerns and increase your confidence in the paper. As you will see, most of your concerns regarding lack of information were actually left out of our paper due to space constraints.
>
> W1: A deep learning model (e.g., SAM) did not generate our annotations. In section 3.1, we briefly described our annotation process. Unfortunately, page limitations prevented us from providing a thorough explanation, so as for your concern, we added an abridged version of the following description to the main content. We have used V7's Darwin automated annotation tool (https://www.v7labs.com/) with double-independent checking in our quality control process. We do not have a commercial or any other relationship with V7, and they were selected after examining several tools. Even though it is not open source but with a paid plan, it was chosen for several reasons, which have now been added to the revised version, section 3.1
>
>
> W2: Section 3, page 4, describes the vertical trajectories. In the training and validation set, there are 2175 images, and only one out of five images were taken from the training set. Regarding the table, we believe your remark is correct, but unfortunately, we could not add a Table to the main content of the paper due to space constraints. We have now added a table similar to Table 1 to the Appendix. Thanks you for your comment!
>
> W3:  The validation set includes horizontal trajectories from the Agamim area. These trajectories partly overlap with the vertical ones, yielding a slight distributional shift. This shift prevents overfitting when calibrating the hyper-parameters if such an optional step is performed. As was mentioned in 3.2, we combined the training and the validation sets when we trained our models to obtain a larger and more diverse set, thus obtaining better models. As for the test set, there is a more profound distributional shift as it was captured in a different neighborhood. This makes the test set more challenging and better tests the model’s generalization ability.
>
> W4: We need to apologize for misleading your interpretation. The last paragraph of Section 4.1 is unclear and refers to the experiment presented in Figure 7. We have now revised the text and hope it is much clearer.
>
> W5: The roof of the building was not defined as a place safe for landing, only ground locations. Therefore, the roof and all objects on it were included in the class “building,” even though there were buildings with different objects that matched our annotation taxonomy located on the roof. Although the ground truth did not include labeled objects on the roof, the model classified those objects. This gave rise to a misclassification error when calculating the performance according to the ground truth. From a utility point of view, this error can be removed; hence, we also calculated the performance with no buildings. In W1, we described our rigorous annotation process. We have worked hard to correct errors in the annotations, and to the best of our knowledge, there are no errors in the dataset; however, if any errors are found, we will be happy to correct them.
>
> R1: Removed
>
> R2: Corrected
>
> R3: We have highlighted Figure 7 in our third main contribution in the Introduction.
>
> R4: Sharing all of our data will indirectly disclose our names, violating the journal's strict double-blind policy. For this reason, we have shared a reduced version of our data at https://github.com/messi-dataset/. Since we aim to share Messi with the broadest possible audience, code and data will become available to the community once the paper is published.
>
> R5: This is another very helpful remark connected with W4. Thank you! In the revised version, Section 4.2, we have added to the discussion of Figure the model used (“SegFormer-B3”) and the corresponding weighting scheme (“sqrt”). The last paragraph of section 4.1 was revised accordingly.

---

### Review · Reviewer_sAFp · 2025-01-03

**Summary Of Contributions:**

This paper introduces the Multi-Elevation Semantic Segmentation Image (MESSI) dataset, which consists of images captured by a drone flying over four distinct urban scenes at varying altitudes. The study presents the dataset and evaluates multiple state-of-the-art semantic segmentation methods using it.

**Audience:**

Yes

**Broader Impact Concerns:**

The paper introduces a dataset consisting of images captured by a drone flying over four different urban scenes at varying altitudes. This requires a clearer motivation and an ethical statement, as the dataset could potentially be misused for military research.

**Claims And Evidence:**

No

**Requested Changes:**

The work needs to be carefully revised. The authors should address the inconsistent annotations present on the roofs of buildings. Additionally, the presentation, and particularly the writing, needs improvement. This should begin with a clear and well-structured motivation in the introduction section. The authors must also explain in detail how the dataset was created, including how images are aligned across different altitudes. Furthermore, the paper needs to propose a clear methodology for assessing the influence of altitude on the segmentation performance of different methods.

**Strengths And Weaknesses:**

## Strengths:

S1: The paper introduces an innovative and practical concept by providing real-world data of the same scene captured at different altitudes, potentially offering valuable insights for in depth analysis.

## Weaknesses:

W1: The paper claims multiple times that the MESSI dataset enables the study of the effect of images taken from different altitudes on semantic segmentation algorithms, a property that is hard to benchmark using existing datasets. However, to analyze these effects in depth, the frames need to be aligned. Given Image 1 at an altitude of 30m and Image 2 at 50m, one would need the transformations to determine which areas in Image 2 are covered in Image 1, assuming a quasi-static scene. Analyzing class IoUs or mIoU for different altitudes is a first step but cannot provide in-depth insights. While this is mentioned briefly on page 10, it is not described in detail in the paper. The authors should provide more insights into this, as it is the foundation of the work. More importantly, this feature is barely utilized in the analysis. What is the detailed influence of having data at different flight altitudes on performance?

W2: The authors realized throughout the process that the "building" class is “falsely annotated” for rooftops with other objects on top, as shown in the example in Fig. 5. The proposed solution is to simply ignore the building class: “This problem can be somewhat relaxed by ignoring the Building class both in training and testing, and therefore, each experiment was repeated with that category omitted.” To summarize, the dataset is poorly annotated for a major class in urban scenarios, and the solution to ignore the class is inadequate.

W3: The paper presents an image dataset. The data and concept of the dataset should be visualized by showing the images, the alignment based on drone altitude, and the ground truth in the main paper, rather than only in the supplemental material.

W4: The writing is rough and partly unscientific. For example, phrases like “the nice review” (Introduction, second paragraph, second sentence) are overly informal. The paper would benefit from clearer writing and a more prominent storyline. In particular, the introduction would benefit from a clearer motivation and narrative. For example, the novel aspects of the dataset and the rationale for having data for the same location at different altitudes are hidden in the text and should be more prominent. Additionally, the overview of the paper’s sections is unnecessary in a research paper and could be omitted.

## Minor Remarks:
- Including the footnote text in the OpenReview abstract is confusing.
- Abstract: "can be used … for semantic segmentation or other applications of interest" — this is not scientific. Which specific applications can be addressed with the provided data?
- Sec. 2, first sentence: "The motivation for building the MESSI was to provide data to train and baseline deep-learning semantic segmentation algorithms" — should this be "the MESSI dataset"? Also, what does it mean to "baseline deep-learning semantic segmentation algorithms"?
- Table 1: Formatting is broken. The word “Coverage” overlaps with the table content.
- Figure 2: Could be visualized in log scale, with bars colored according to their respective class colors in the annotations (see [A]).
- Include the number of images in the train, validation, and test splits.
- Fig. 4: Difficult to interpret. What is shown on the x-axis?
- Should be "Cityscapes" instead of "CityScapes" [A].
- Beginning of page 6: "each image was divided into 3×3 of 2048 by 1366 overlapped tiles" — this is not a complete sentence.
- Fig. 6: Numbers are hard to read due to suboptimal coloring.
- Sec. 5: "This paper presents a new dataset called 'MESSI,' designed to serve as a baseline for semantic segmentation algorithms." — how can a dataset serve as a baseline?
- Fig. 7: Appears in the middle of the abstract.
- References: Inconsistent formatting, e.g., "Cordts et al. In Proceedings of the IEEE conference on computer vision and pattern recognition" vs. "Geiger et al., In Proc. of the IEEE Conf. on Computer Vision and Pattern Recognition (CVPR)." Venue names are inconsistently abbreviated. Additionally, capitalization in paper titles is inconsistent.

[A] Marius Cordts, Mohamed Omran, Sebastian Ramos, Timo Scharwächter, Markus Enzweiler, Rodrigo Benenson, Uwe Franke, Stefan Roth, and Bernt Schiele. The Cityscapes dataset for semantic urban scene understanding. In CVPR, pp. 3213–3223, 2016.

---

> ### Author Response · Authors · 2025-01-08
>
> Dear Reviewer sAFp:
> Thank you for your comprehensive and in-depth review. We have uploaded a revised version of the paper based on your remarks. We would like to address your concerns and hopefully clarify them.
>
> W1: Thank you for appreciating the importance of our claim. However, our paper aims to present MESSI to the research community, and because we have page limitations, we tried to show different analysis aspects of the dataset. Figure 7 emphasizes the importance of an elevation-based dataset. Shortly before this review, we revised our paper and added more details in the last paragraph of section 4.1 and on the analysis of Figure 7 (e.g., model and weighting scheme).
> A strong claim in your review is that frames need to be aligned. As opposed to this, we wanted to share raw data gathered by our drone and enough data (precise pose, intrinsic parameters) to allow users to use the material as they feel more appropriate (including aligning frames). Moreover, as you stated in your introduction, we aimed to make MESSI available to the community and not show all possible applications. As such, we presented an analysis example and are currently working on several different extensions. Please note that space limits made us focus on our main purpose: to share MESSI with the community.
>
> W2: We apologize for misleading you in your interpretation; the first review we received has the same claim!  The initial motivation for our research, for which MESSI is one of the building blocks, is to find a landing spot in an urban environment for an aerial taxi. As we advanced, we found other applications that could benefit from our research, including forced/emergency landing and survival extraction. Consequently, in the main body, we referenced Section 3 in the Appendix, explaining that the segmentation taxonomy was designed to find safe, obstacle-free ground areas suitable for landing while considering human, property, and drone safety. That is why rooftops were not defined as safe places to land but rather only ground locations. Hence, the roof and all objects on it were, by design, included in the class "building," even though there were buildings with different objects that matched our annotation taxonomy located on the roof. Although the ground truth did not include labeled objects on the roof, the model classified those objects. This gave rise to a misclassification error when calculating the performance according to the ground truth. This is not a "false annotation" but rather a design decision based on the problem we wanted to solve. From a utility point of view, this "error" can be removed; hence, we also calculated the performance with no buildings. As you probably noticed, large numbers of objects were annotated, especially from higher altitudes. We have worked hard to correct errors in the annotations, and to the best of our knowledge, there are no errors (or false annotations) in the dataset; however, if any errors are found, we will be happy to correct them.
>
> W3: You are correct in principle, but unfortunately space limitations constrained us. We tried to describe the dataset's characteristics and analysis in the main paper and give visualization examples and supplementary information in the Appendix. In this respect, we followed the example from other similar papers including Cityscapes.
>
> W4: Thank you. The overview of the paper's sections was omitted. "nice review" was deleted.
>
> MR: We appreciate your thorough reading. Some of your remarks were correct in the previous revised version (e.g., Table 1: Formatting is broken), and others are corrected in the new revised version. Regarding "other specific applications that can be addressed with the provided data": they are mentioned in Section 1, main contribution, bullet 2. Thank you. We have now added them to the Abstract. We hope we have clarified your concerns. We appreciate your minor remarks. Most of them were considered to improve our paper:
>
> We removed the footnote.
>
> Improve our Abstract.
>
> Corrected to "MESSI dataset" and replaced "baseline" with "compare."
>
> Corrected Table 1 formatting.
>
> Fig. 2 is updated. We tried the log scale but did not get informative results. Thank you for this remark!
>
> Fig. 4 is updated. The x-axis is now coherent with the rest of the figures.
>
> Corrected Cityscapes.
>
> Fig. 6 colormap is changed.
>
> Thank you. Corrected to validation benchmark.
>
> We are confused. Figure 7 is far from the Abstract.
>
> Made references consistent. Thank you for noticing!
>
> RC: We reviewed our paper per your suggestions and introduced several changes based on your remarks. In particular, we have added a description of the annotation process to Section 3.1, modified figures, titles, etc'. Overall, clarity has improved, and we thank you for that. We stress that the paper's main objective is to share MESSI with the community.
>
> BIC: We appreciate your concern. Our research started and was originally funded by a private company planning an aerial taxi service.

---

### Review · Reviewer_GNHy · 2025-05-01

**Summary Of Contributions:**

The paper introduces a new dataset called MESSI, taken by a drone in an urban environment. MSSEI contains images from various altitudes and has a large class imbalance, which can serve as a challenging evaluation benchmark for semantic segmentation algorithms.  Classical segmentation models and three different weighting policies have been evaluated on MESSI.

**Audience:**

Yes

**Claims And Evidence:**

Yes

**Requested Changes:**

1.	The realized works section needs to be reorganized. It is better to introduce related work in a certain order, such as publication date or relevance. Alternatively, the deficiencies of the current datasets can be categorized and summarized into several categories, and finally, it would be claimed that this work has overcome or compensated for these deficiencies.
2.	Some expressions are unclear and confusing:
   (1) In “To mention two”, what does “two” refer to?  As there are several datasets following this sentence.
   (2) Similarly, in “Notice that these three datasets deal with”, what does “these three” refer to?
   (3) “Further motivation for this problem may also be found in Pinkovich et al. (2022)”. It is better to explain the motivations in this paper, rather than let the readers find them in other papers.
   (4) In “models;and 3) almost unanimously”, there is a missing space.
   (5) The model name of SegFormer-B3 should be fixed, instead of sometimes “SegFormer-B3” and sometimes “Segformer-B3”.
3.	Missing out on some crucial content and details.
(1) In Figure 1, “Left: the trajectory (red)”, but there is no red instead of orange in the left subfigure. In addition, why is the view of “Ha-Medinah Square” absent? It is better to add it in Figure 1.
(2) In Page 5, as the paper states that Ir Yamim and Ha-Medinah Square are test set and Paths A to C are the validation set, what is the composition of training set?  It is suggested to draw a table to clearly explain how to divide these sets and the number of samples for each set, so as to form a unified standard for the latter researchers to evaluate their models.
4.	Please explain what has been done to deal with the domain shift in Ha-Medinah Square, especially when it is a part of the test set.
5.	Please explain why the mIoU of Mask2Former-Swin-B becomes lower using Equal and Sqrt, meanwhile its mAcc w/o Building is higher than the mAcc with all categories in Table 2.
6.	Please explain why the mIoU of Segformer-B3 in Table 2 is inconsistent with that in Table 3.

**Strengths And Weaknesses:**

## Strengths

1. A new dataset is introduced for evaluating semantic segmentation algorithms.  It contains the visually rich images captured by a drone flying over several areas in different horizontal and vertical trajectories. This would be of interest to several people in the TMLR community.
2. Recent SOTA segmentation models with three different weighting policies have been tested on MSSEI.
3. This paper is easy to read and uses many tables and figures for an intuitive visualization.

## Weaknesses

1. The real work section is quite chaotic and lacks clear organization. The related works should be presented in a clear sequence, either by publication date or relevance to your work.
2. Some expressions are unclear and confusing.
3. Missing out on some crucial content and details, such as the composition of the training set on Page 5 and the view of “Ha-Medinah Square” in Figure 1.
4. The papers mentioned that “The Ha-Medinah Square sequence has a considerable domain shift”, but the papers did not explain how to overcome the domain shift in Ha-Medinah Square. If possible, domain shift can also be used as a research direction for this dataset, so as to increase the contributions of this work.
5. When discarding the Building category, the mIoU of models in Table 2 normally increases, but the mIoU of Mask2Former-Swin-B becomes lower using Equal and Sqrt, meanwhile, its mAcc w/o Building is higher than the mAcc with all categories. Why does this happen?
6. Why is the mIoU of Segformer-B3 in Table 2 inconsistent with that in Table 3?

---

> ### Author Response · Authors · 2025-05-06
> **Answer to Reviewer GNHy**
>
> Dear GNHy:
>
> Thank you for your comprehensive and in-depth review. We have uploaded a revised version of the paper based on your remarks. We would like to address your concerns and hopefully clarify them.
>
> RC 1:
> We agree with the reviewer about the lack of clear organization. We tried several different alternatives but were not satisfied with any of them. Eventually, we included a table summarizing each reference and allowing a comparison with MESSI. The more detailed discussion is still included.
>
> RC2:
>
>     1,2,3. Thank you for your remark. Paragraph rephrased.
> 	4. Corrected.
> 	5. Thank you for noticing, Model names were changed to SegFormer-B3.
>
> RC3:
>
> 	1. “Ha-Medinah Square” was added. Red was changed to Orange.
> 	2. A table that summarizes the chosen composition of the training, validation, and test sets was added
>
> RC4:
> Addressing the domain shift in Ha-Medinah Square, particularly as part of the test set, is an important consideration. In this paper, our primary focus is the introduction and evaluation of MESSI as a novel dataset. While we did not specifically address domain shift, we recognize its significance and intend to explore effective mitigation strategies in our future research.
>
> RC5:
> Mean Intersection over Union (mIoU) metric can be a biased measure, particularly when evaluating performance on datasets with significant class imbalance. This bias occurs because mIoU calculates the average IoU across all classes. Consequently, a low IoU score on a single, often small or rare, class can disproportionately lower the overall mIoU, even if the model performs well on larger, more frequent classes.
>
> As reported on page 8, section 4.2, in Wang et al. 2023 and shown in the Mask2Former performance table in the appendix, Mask2Former performed significantly worse on smaller, less frequent classes, directly contributing to a lower overall mIoU score due to the metric's inherent bias.
>
> To counteract this issue and improve performance on underrepresented classes, we experimented with different weighting policies during model training:
>
> 1. Equal Weighting: This method assigns the same weight to every class, regardless of its frequency in the dataset. It serves as a baseline but does not address the class imbalance problem. Therefore, it did not improve segmentation results for smaller classes, as they did not receive increased emphasis during training. When the very frequent "building" class was removed, the total number of pixels across the remaining classes decreased, and the weights were then re-normalized based on this new distribution. As a result, the relative weight assigned to the remaining frequent classes increased. Further reducing the focus on the already rare classes within this subset, leading to even poorer performance for Mask2Former.
>
> 2. Square Root (Sqrt) Weighting: This method applies a weight inversely proportional to the square root of each class's frequency. While it does give more weight to rare classes than equal weighting, the square root function makes it a "softer" adjustment and not aggressive enough to overcome the dataset imbalance on its own fully. Like equal weighting, removing the "building" class, and recalculating weights, the sqrt weighting scheme still did not assign sufficient weight to the rare classes relative to the remaining frequent ones, again resulting in poorer performance for Mask2Former on those classes.
>
> 3. Proportional Weighting (Inverse Frequency): This scheme assigns a weight inversely proportional to the frequency of each class in the dataset. This method directly and strongly compensates for class imbalance by giving significantly higher weight to rare classes. This strategy proved effective for Mask2Former. In particular, when removing the dominant "building" class, the proportional weighting scheme ensured that rare classes were no longer ignored, leading to significantly higher or at least equal segmentation scores for these classes (as detailed in the appendix table), demonstrating that proportional weighting successfully enabled the model to overcome the challenges posed by class imbalance.
>
> RC6:
> Thank you for pointing this out. Table 3 provides detailed results for all the classes without the "water" class; consequently, the mIoU is calculated for all classes without the "water" class. The "water" class is one of the rarest classes, and the models' segmentation results are close to zero for all the weighting methods, biasing the overall results. Regrettably, we forgot to point this out in the SegFormer table and the Mask2Former table in the appendix. We have now added the water class and the mIoU results with and without the water to the tables. Thank you for noticing. We would also like to point out that due to your remark, we checked that all the values in the tables were copied correctly from our raw results and found a couple of entries that we did not copy correctly. Again, we have to thank you for this.

---

### Author Response · Authors · 2025-01-28
**Concerns and clarifications**

Dear all,

If you have any more concerns or need more clarification, we will happily address them.

---

### Author Response · Authors · 2025-08-06
**Camera Ready Revision**

A Camera Ready Revision was uploaded. A link for the full dataset was shared in the manuscript, and a link for the code was shared here. We thank the action editors and the reviewers for their in-depth reviews.

---

### Decision · Action_Editor_PkWo · 2025-07-19

**Recommendation:** Accept with minor revision

**Audience:**

Yes

**Audience Explanation:**

MESSI fills a gap in current datasets by offering real-world urban scenes captured at multiple altitudes, which is relevant to tasks such as autonomous navigation, semantic segmentation under varying viewpoints, and domain adaptation.

The proposed dataset is relevant to both applied and algorithmic ML research. Applications are clearly explained in the paper and any aerial vision system research would benefit from the paper. Moreover, the paper’s focus on class imbalance, altitude variation, and generalization to out-of-distribution test sets aligns well with topics of interest in the algorithmic ML community.

**Claims And Evidence:**

Yes

**Claims Explanation:**

The paper is reviewed by 3 reviewers and all reviewers recommended acceptance.

The main claims of the paper are: 1) The introduction of a new drone-based semantic segmentation dataset (MESSI) with multi-altitude images.
2) That this dataset is useful for benchmarking segmentation algorithms under class imbalance and varying elevation. 3) That baseline experiments reveal useful insights about weighting schemes and segmentation performance across altitudes.

These claims are well-supported. The dataset is novel in its use of multi-altitude data and is accompanied by meaningful metadata. The experiments are extensive and include various models and weighting schemes.